# Cancer Mutations in FGFR2 Prevent a Negative Feedback Loop Mediated by the ERK1/2 Pathway

**DOI:** 10.3390/cells8060518

**Published:** 2019-05-29

**Authors:** Patrycja Szybowska, Michal Kostas, Jørgen Wesche, Antoni Wiedlocha, Ellen Margrethe Haugsten

**Affiliations:** 1Department of Molecular Cell Biology, Institute for Cancer Research, The Norwegian Radium Hospital, Oslo University Hospital, Montebello, 0379 Oslo, Norway; Patrycja.Szybowska@rr-research.no (P.S.); Antoni.Wiedlocha@rr-research.no (A.W.); 2Centre for Cancer Cell Reprogramming, Institute of Clinical Medicine, Faculty of Medicine, University of Oslo, Montebello, 0379 Oslo, Norway; Michal.Janusz.Kostas@rr-research.no (M.K.); Jorgen.Wesche@rr-research.no (J.W.); 3Department of Tumor Biology, Institute for Cancer Research, The Norwegian Radium Hospital, Oslo University Hospital, Montebello, 0379 Oslo, Norway; 4Military Institute of Hygiene and Epidemiology, 01-163 Warsaw, Poland

**Keywords:** FGFR2, ERK1/2, phosphorylation, serine, negative feedback loop, cancer

## Abstract

Tight regulation of signaling from receptor tyrosine kinases is required for normal cellular functions and uncontrolled signaling can lead to cancer. Fibroblast growth factor receptor 2 (FGFR2) is a receptor tyrosine kinase that induces proliferation and migration. Deregulation of FGFR2 contributes to tumor progression and activating mutations in FGFR2 are found in several types of cancer. Here, we identified a negative feedback loop regulating FGFR2 signaling. FGFR2 stimulates the Ras/MAPK signaling pathway consisting of Ras-Raf-MEK1/2-ERK1/2. Inhibition of this pathway using a MEK1/2 inhibitor increased FGFR2 signaling. The putative ERK1/2 phosphorylation site at serine 780 (S780) in FGFR2 corresponds to serine 777 in FGFR1 which is directly phosphorylated by ERK1/2. Substitution of S780 in FGFR2 to an alanine also increased signaling. Truncated forms of FGFR2 lacking the C-terminal tail, including S780, have been identified in cancer and S780 has been found mutated to leucine in bladder cancer. Substituting S780 in FGFR2 with leucine increased FGFR2 signaling. Importantly, cells expressing these mutated versions of S780 migrated faster than cells expressing wild-type FGFR2. Thus, ERK1/2-mediated phosphorylation of S780 in FGFR2 constitutes a negative feedback loop and inactivation of this feedback loop in cancer cells causes hyperactivation of FGFR2 signaling, which may result in increased invasive properties.

## 1. Introduction

Tight regulation of receptor tyrosine kinase signaling is required for specific cellular responses, such as cell growth, differentiation, migration, and apoptosis. Inadequate regulation of signaling is a common event in cancer development and enhanced receptor signaling promotes tumor growth [1]. The receptor tyrosine kinase, FGFR2 (fibroblast growth factor receptor 2) is a transmembrane, cell-surface localized receptor that belongs to a family of four related receptors [2]. FGFR2 is activated by FGF ligands and induces various downstream signaling molecules. Deregulation of FGFR2 contributes to tumor progression and activating mutations in FGFR2 have been found in different types of cancer, like gastric cancer, breast cancer, and endometrial carcinoma [3,4]. In addition, activating mutations have been found in skeletal disorders, like Apert syndrome and Crouzon syndrome [5]. Clearly, precise regulation of FGFR2 signaling is important to prevent diseases.

Upon ligand binding, FGFRs dimerize. This, in turn, activates the tyrosine kinase domain of the receptor by trans-autophosphorylation [2]. FGFRs mediate signaling by recruiting specific molecules that bind to phosphorylated tyrosines, triggering a number of signaling pathways. The docking protein FRS2 (FGFR substrate 2) is phosphorylated by the activated receptor, creating phosphotyrosine docking sites for proteins containing SH2-domains. By binding to FRS2, the adaptor protein Grb2 (growth factor receptor-bound protein 2) activates the Ras/ mitogen-activated protein kinase (MAPK) pathway and the phosphoinositide 3-kinase (PI3K)/Akt pathway [2]. Ras activates the kinase activity of Raf, which phosphorylates MEK1/2. MEK then phosphorylates ERK1/2 (extracellular signal-regulated kinase) which activates 90 kDa Ribosomal S6 Kinase 2 (RSK2), among other downstream targets. Activated FGFRs also recruit and phosphorylate phospholipase Cγ (PLCγ), which culminates in the activation of protein kinase C (PKC) [2].

In comparison to the well-studied activation of FGFRs, the mechanisms leading to deactivation of the receptor are not fully understood. It is known that the signal from the activated receptor can be attenuated by internalization and degradation in lysosomes [6,7]. After internalization, FGFR ubiquitination marks the receptor for degradation [6,8]. Depending on the receptor type, the bound ligand, and possibly also the cell type and the context, FGFRs might also be recycled back to the cell surface instead of being transported to lysosomes, which may result in prolonged signaling [7].

FGFR signaling is also regulated by phosphatases. Recently, we have shown that a phosphatase, PTPRG, directly dephosphorylates activated FGFRs [9]. Proteins that regulate FGFR signaling, such as MAPK phosphatase 3 (MKP3) and Sprouty 1/2, are negative regulators that are induced or activated by FGF signaling and act on downstream signaling molecules [10]. In addition, FGFR signaling can be regulated by inhibitory phosphorylation, forming negative feedback loops that attenuate the signals. It has been shown that active ERK1/2 can phosphorylate FRS2 on threonine residues. This leads to reduced tyrosine phosphorylation of FRS2 and therefore reduced downstream signaling [11]. On the receptor level, two such negative feedback loops have been identified for FGFR1 [12,13]. It has been shown that upon FGFR1 activation/tyrosine phosphorylation, the receptor is also phosphorylated at serine 777 (S777) directly by activated ERK1/2. S777 phosphorylation reduces the tyrosine phosphorylation in the kinase domain of the receptor and thus also reduces signaling [12]. In addition, the serine/threonine kinase RSK2, which is activated through the Ras-MAPK pathway, can also bind to FGFR1 and phosphorylate FGFR1 at serine 789 [13]. This phosphorylation seems to be required for proper endocytosis and ubiquitination of FGFR1. Preventing RSK2 activation or mutation of S789 leads to increased signaling [13]. It is not clear if the other FGFRs are also regulated by such negative feedback loops.

Here, we have investigated whether a similar negative feedback loop mediated by ERK1/2 also exists for FGFR2. Inhibition of the ERK1/2 signaling pathway, using a MEK1/2 inhibitor (U0126), led to sustained FGFR2 phosphorylation. Moreover, substitution of serine 780 (S780) in FGFR2 for alanine also resulted in sustained FGFR2 activation. S780 in FGFR2 is equivalent to the ERK1/2 substrate S777 in FGFR1. Several truncated forms of FGFR2 lacking the C-terminal tail, including S780, have been identified in cancer. In addition, S780 has been found mutated to leucine in a patient with bladder cancer. Substituting S780 in FGFR2 with leucine also increased FGFR2 signaling. More importantly, cells expressing the mutated versions of S780 were migrating faster than cells expressing wild-type FGFR2. Possibly, the lack of MAPK-dependent negative feedback gives FGFR2-expressing cancer cells an advantage. These results also indicate that care should be taken when the MAPK-pathway is inhibited in cancer. 

## 2. Materials and Methods

### 2.1. Materials, Antibodies, and Compounds

The following antibodies were used: Mouse anti-phospho-ERK1/2 (Thr202/Tyr204) (#9106), rabbit anti-ERK1/2 (#9102), mouse anti-phospho-FGFR (Tyr653/654) (#3476), rabbit anti-FGFR2 (#11835), rabbit anti-FGFR2 (N-terminal) (#23328), rabbit anti-FGFR1 (#9749), rabbit anti-FGFR3 (#4574), rabbit anti-FGFR4 (#8562), rabbit anti-phospho-PLCγ (Tyr783) (#14008), and rabbit anti-phospho-RSK2 (Ser 227) (#3556) from Cell Signaling Technology (Leiden, The Netherlands) and mouse anti-γ-tubulin (T6557) from Sigma-Aldrich (St. Louis, MO, USA). Fluorescently labelled secondary antibodies were from Jackson ImmunoResearch Laboratories (Cambridgeshire, UK). HRP-conjugated secondary antibodies were from Jackson ImmunoResearch Laboratories and Agilent (Santa Clara, CA, USA).

U0126 (1144) was from Tocris Bioscience (Bristol, UK). PD173074 was from Calbiochem (San Diego, CA, USA). Cycloheximide, recombinant EGF, mowiol, heparin, and protein-G-sepharose were from Sigma Aldrich. Restriction enzymes were from New England Biolabs (Ipswich, MA, USA). Adenosine triphosphate [γ-^32^P] 3000 Ci/mmol EasyTides was purchased from PerkinElmer (Norwalk, CT, USA). PhosSTOP phosphatase inhibitor cocktail and cOmplete EDTA-free protease inhibitor cocktail were from Roche (Basel, Switzerland). Hoechst 33342, DyLight 550 NHS Ester, and recombinant active ERK1 with glutathione S-transferase (GST) tag (#PV3311) were from Thermo Fisher Scientific (Waltham, MA, USA). Recombinant FGF1 was prepared as previously described [14]. FGF1 was labelled with DyLight 550 (DL550-FGF1) following the manufacturer’s procedures.

### 2.2. Plamids and siRNAs

cDNA encoding full-length human FGFR2 (IIIc) (NCBI: NM_000141) was cut out from the pCMV6-XL4 cDNA clone (Origene Technologies, Rockville, MD, USA) as an *EcoRI-XbaI* fragment and ligated into pcDNA3 (Thermo Fisher Scientific, Waltham, MA, USA). The resulting plasmid was further cut with *KpnI* to remove the upstream untranslated region. To remove the untranslated region downstream of the gene, the plasmid was partially cut with *Tth111I,* followed by cutting with *XbaI*. The plasmid was furthermore treated with T4 DNA polymerase (New England Biolabs, Ipswich, MA, USA) to make blunt ends and then ligated. Note that the *XbaI* and the *Tth111I* sites were destroyed. After sequencing, a point mutation in the N-terminal region was discovered (G183V). This point mutation was mutated back (generating a glycine at the 138 position) using site-directed mutagenesis with the following primer: 5-CGCTGCCCAGCCGGGGGGAACCCAATGCCAACC-3. pcDNA3 hFGFR2 was used as a template to generate pcDNA3 hFGFR2 S780A, S780D, and S780L. The following primers were used: S780A; 5-CCTCTCGAACAGTATGCACCTAGTTACCCTGAC-3, S780D; 5-CCTCTCGAACAGTATGACCCTAGTTACCCTGAC-3, S780L; and 5-CCTCTCGAACAGTATCTACCTAGTTACCCTGAC-3. All constructs were verified by sequencing (Eurofins Genomics, Ebersberg, Germany). pcDNA3 hFGFR1 and pcDNA3 hFGFR4 have been described previously [7,15] and pcDNA3 hFGFR3 was a generous gift from Dr. A. Yayon (ProChon Biotech, Ness Ziona, Israel).

### 2.3. Cell Lines and Transfection

To generate U2OS cells stably expressing FGFR2, FGFR2 S780A, FGFR2 S780D, and FGFR2 S780L, Fugene 6 transfection reagent (Promega, Madison, WI, USA) was used according to the manufacturer’s protocol. Clones were selected with 1 mg/mL geneticin and then the clones were chosen based on their receptor expression levels analyzed by immunofluorescence and Western blotting. Throughout the paper, clone #1 of the particular stable cell line is used if nothing else is stated.

The cells were propagated in Dulbecco’s Modified Eagle Medium (DMEM) supplemented with 10% fetal bovine serum, 100 U/mL penicillin, and 100 μg/mL streptomycin in a 5% CO_2_ atmosphere at 37 °C.

Transient transfection was performed using Fugene 6 transfection reagent according to the manufacturer’s protocol. Cells were analyzed 16–24 h after transfection.

### 2.4. Western Blotting

Cells were treated as indicated and then lysed in Laemmli sample buffer (Bio-Rad, Oxford, UK). Proteins in the cell lysates were separated on a gradient (4–20%) sodium dodecyl sulfate-polyacrylamide gel electrophoresis (SDS-PAGE) and then blotted onto a membrane using the TransBlot^®^ Turbo Transfer system (Bio-Rad). Membranes were then incubated with indicated primary antibodies followed by corresponding secondary antibody coupled to HRP. Bands were visualized by chemiluminscence using SuperSignal™ West Dura Extended Duration Substrate (Thermo Fisher Scientific, Waltham, MA, USA) or SuperSignal™ West Femto Maximum Sensitivity Substrate (Thermo Fisher Scientific). In some cases, antibodies were stripped from the membranes using Pierce Stripping buffer and the membranes were reprobed. The images were prepared using ImageLab Software (Bio-Rad) and Adobe Illustrator CS4 14.0.0 (San Jose, CA, USA). Quantification of bands of interest was performed in Fiji ImageJ software [16]. Lane normalization factor (LNF) was determined by dividing the intensity of the γ-tubulin bands on its highest signal in each blot. 

### 2.5. Microscopy

Cells, seeded onto coverslips, were treated as indicated and fixed in 4% formaldehyde. The cells were then permeabilized with 0.1% triton X-100, stained with indicated antibodies and Hoechst 33342 and mounted in mowiol. Confocal images were acquired with a 63X objective on a Zeiss confocal Laser Scanning Miscroscope (LSM) 780 (Jena, Germany). Images were prepared in Fiji Image J software and Adobe Illustrator CS4 14.0.0. Images for quantification of p-FGFR and DL550-FGF1 signal intensities were taken with identical settings and the quantification was performed with Fiji Image J software. The same threshold was used for all images in the same experiment. Due to background staining in the nuclei, p-FGFR intensities in the nuclei were subtracted from the total intensities in the corresponding cell. 

### 2.6. In Vitro Phosphorylation Assay

The cells were starved for 2 h in serum-free media and lysed in lysis buffer (20 mM phosphate-Na pH 7.4, 150 mM NaCl, 1 mM Ethylenediaminetetraacetic acid (EDTA), 1% Triton X-100, protease inhibitors). The receptors were immunoprecipitated for 1 h using anti-N-terminal-FGFR2 antibodies pre-bound to protein-G-sepharose, washed 3 times with 1 M NaCl and treated with 1 µM PD173074 for 30 min. The kinase reaction was performed on beads using 50 ng recombinant active ERK1 and 50 µCi ATP-γ-^32^P (per 100 µL reaction) in 50 mM HEPES-Na pH 7.5, 20 mM MgCl_2_, 5 mM Ethylene Glycol Tetraacetic Acid (EGTA), and phosphatase inhibitors for 30 min at 30 °C. The reaction was quenched with 20 mM EDTA. Then, the immunoprecipitated receptors were washed 3 times (25 mM HEPES-Na pH 7.5, 1 mM EDTA) and released from the beads in SDS-loading buffer by 15 min at 95 °C and subjected to SDS-PAGE before analysis with autoradiography and immunoblotting.

### 2.7. Cell Migration

Cells sparsely seeded in IncuCyte Image Lock 96-well plates (Essen BioSciences, Hertfordshire, UK) were imaged every 10 min for 21 h by IncuCyte^®^ S3 Live Cell Analysis System with IncuCyte^®^ S3 Software (V2018B) (Essen BioSciences). In all experiments, cells were either left untreated or treated with FGF1 (100 ng/mL) and heparin (20 U/mL). Images were analyzed with IncuCyte^®^ S3 Software (V2018B) and Fiji ImageJ software with Manual Tracking and Chemotaxis and Migration Tool (ibidi GmbH, Planegg, Germany). 

## 3. Results

### 3.1. Inhibition of MEK1/2 Increases FGFR2 Signaling

Signaling from FGFRs is regulated by mechanisms such as endocytic trafficking [6,7] and dephosphorylation by phosphatases (PTPRG) [9]. Recently, we identified a negative feedback loop that involves direct phosphorylation of serine 777 (S777) in the C-terminal tail of FGFR1 by active ERK1/2 [12]. Phosphorylation of S777 in FGFR1 is necessary for proper attenuation of FGFR1 signaling and treatment of cells with U0126, a MEK1/2 inhibitor, leads to increased activation of FGFR1. To investigate if a similar ERK1/2-mediated negative feedback loop also exists for FGFR2, we treated cells with U0126 MEK1/2 inhibitor and investigated tyrosine phosphorylation status of FGFR2 at different time-points after addition of FGF1. Since FGFR levels are low in many cells and endogenous FGFRs can be difficult to detect, we generated U2OS cells stably expressing FGFR2 IIIc (U2OS-R2). In contrast to parental U2OS cells, our U2OS-R2 cells endocytose detectable amounts of DL550-FGF1 (FGF1 labelled with DyLight550) and are strongly stained with anti-FGFR2 antibodies (Figure 1a). U2OS cells do not express detectable levels of any of the four FGFRs (Appendix A and [17]) and, although the antibody that we use against phosphorylated FGFR (p-FGFR antibody) recognizes all four receptors, only the ectopic FGFR in the stably transfected U2OS cells is detected.

First, we investigated which doses of the MEK1/2 inhibitor (U0126) efficiently inhibit ERK1/2 activation upon FGF1 stimulation in U2OS-R2 cells (Figure 1b). MEK1/2 is upstream of ERK1/2 in the Ras/MAPK signaling pathway. Incubating the cells with increasing concentrations of U0126 demonstrated that 20 µM U0126 efficiently blocked ERK1/2 activation. Next, we treated the cells with 20 µM U0126 and compared the receptor activation in treated cells versus untreated cells. The levels of tyrosine-phosphorylated FGFR2 were increased in U0126 treated cells compared to untreated cells (Figure 1c). Similar effects were observed in two additional clones of U2OS-R2 (Appendix A). To investigate this effect further, we also stained cells with antibodies against tyrosine phosphorylated FGFR (p-FGFR) and compared the intensity of p-FGFR staining between indicated treatments (Figure 1d). When resting cells or cells treated with FGF1 together with FGFR inhibitor (PD173074) were stained with p-FGFR antibodies, we could detect a bright signal in the nucleus. We considered this as unspecific staining by the antibody. Thus, upon quantification, the intensity of the nuclear p-FGFR antibody staining was subtracted from that of the total cell. Interestingly, in cells treated with FGF1, we observed a clear increase in p-FGFR antibody intensity in the cytosol compared to resting cells or PD173074 treated cells. As expected, we could also observe a high degree of co-localization between DL550-FGF1 and p-FGFR antibody staining (Figure 1d, second panel). When cells were treated with FGF1 and U0126 (to prevent ERK1/2 signaling), we detected an increase in p-FGFR antibody staining compared to FGF1 treatment alone. Taken together, our data indicates that a similar feedback mechanism as to that found for FGFR1 might also exist in the case of FGFR2. We conclude that ERK1/2 signaling is required for attenuation of FGFR2 signaling.

Since ERK1/2 signaling can be activated by other receptor tyrosine kinases as well, we investigated if activation of ERK1/2 prior to FGFR2 activation would influence the response to FGF1. To test this, we treated cells with EGF 30 min prior to stimulation with FGF1 and compared the levels of FGFR2 tyrosine phosphorylation to that in cells not pretreated with EGF. First, we investigated whether EGF activates ERK1/2 signaling in U2OS-R2 cells. We observed a peak of ERK1/2 phosphorylation 10–20 min after addition of EGF. Indeed, ERK1/2 is active in U2OS-R2 cells during this 30 min period of stimulation with EGF (Figure 2a).

Next, we stimulated cells for 30 min with EGF before activation of FGFR by addition of FGF1. Interestingly, reduced levels of tyrosine-phosphorylated FGFR2 was observed in cells pretreated with EGF (Figure 2b). These data indicate a dual role for the ERK1/2 signaling-mediated feedback loop in FGFR2 signaling. Not only does it function to ensure proper attenuation of FGFR2 signaling, it also ensures accurate responses to FGF1 stimulation. In an environment where the ERK1/2 pathway is activated by other receptor tyrosine kinases, the response to FGF1 is less pronounced than in resting cells. In this way, different receptors may cross-talk to prevent excess signaling.

### 3.2. Mutation of Serine 780 in FGFR2 Leads to Increased FGFR2 Activity

Since the phosphorylation site of ERK1/2 in FGFR1 (S777) is already identified, we wanted to investigate if the corresponding serine in FGFR2 is important for proper downregulation of FGFR2 signaling. By sequence alignment, we identified S780 in FGFR2 to correspond to FGFR1 S777 (Figure 3a). Interestingly, in both receptors, the particular serine is followed by a proline and thus forms an ERK1/2 phosphorylation motif (pS/T-P) [18]. We therefore decided to substitute serine 780 in FGFR2 with alanine. Alanine represents a site that cannot be phosphorylated. Next, we prepared U2OS cells stably expressing FGFR2 S780A (U2OS-R2 S780A).

We then investigated whether FGFR2 S780A is expressed to similar levels as the wild-type receptor and if it maintained normal FGFR2 properties. We therefore stimulated cells with FGF1 and analyzed the lysates using Western blotting. First of all, the levels of FGFR2 wild-type and FGFR2 S780A seem comparable in the two clones (Figure 3b). Secondly, we noticed that FGFR2 S780A is able to activate the main downstream signaling pathways similarly to wild-type FGFR2 (Figure 3b). In addition, the mutated receptor was able to bind FGF1 at the cell surface and internalize FGF1 into early endosomes similarly to wild-type FGFR2 (Figure 3c). Comparable results were confirmed in two additional clones of U2OS-R2 wild-type and U2OS-R2 S780A (Appendix A). Moreover, FGFR2 S780A co-localizes with DL550-FGF1, similarly to FGFR2 wild-type (Figure 3d).

We then analyzed the level of FGFR tyrosine phosphorylation over time in FGFR2 S780A-expressing cells. Compared to wild-type expressing cells, FGFR2 activation was sustained in U2OS-R2 S780A (Figure 4a and Appendix A). This effect was similar to the effect observed upon U0126 treatment. It is therefore likely that this serine, also in the case of FGFR2, is phosphorylated by ERK1/2.

In the previous experiments, higher concentrations of FGF1 were used to activate the receptor. We wanted to test if increased FGFR activity also occurred at lower concentrations of FGF1. We treated U2OS-R2 and U2OS-R2 S780A with different concentrations of FGF1 starting at 0.02 ng/mL (Figure 4b). Tyrosine phosphorylation of the receptor and its main downstream signaling pathways were then analyzed with Western blotting. We observed a slight increase in the levels of tyrosine-phosphorylated FGFR2 as well as in the levels of phosphorylated ERK1/2 and PLC-γ in S780A-expressing cells compared to wild-type cells with low concentrations of FGF1. This experiment was performed after 15 min of FGF1 treatment where the effect is not at its highest. However, although the increase in signaling in FGFR2 S780A cells was modest, it was consistent at all concentrations tested. We therefore conclude that the negative feedback loop is operational at both lower and higher concentrations of ligand and at early time points.

Increased signaling can be a result of reduced receptor endocytosis. In Figure 1d, we detected more surface staining and less uptake of DL550-FGF1 in cells treated with U0126 (MEK1/2 inhibitor). This indicates a decrease in endocytosis when MEK1/2-ERK1/2 signaling is inhibited. However, from our previous work on FGFR1, despite a decrease in endocytosis upon MEK1/2-ERK1/2 inhibition, this effect was not due to lack of ERK1/2 phosphorylation of the receptor but rather a lack of a second serine phosphorylation event in FGFR1 mediated by RSK2 [13]. To investigate this, we compared the uptake of DL550-FGF1 in U2OS-R2 wild-type and U2OS-R2 S780A mutant cells in the presence of U0126 (Figure 4c). Upon U0126 treatment, the uptake of DL550-FGF1 was reduced similarly in both cell lines. Thus, the lack of phosphorylation on S780 is probably not the reason for the reduced endocytosis upon MEK1/2 inhibition. Other phosphorylation events mediated by components of the MAPK signaling pathways might be important for proper FGFR2 endocytosis.

Next, we wanted to test if ERK1/2 directly phosphorylates FGFR2. We therefore immunoprecipitated FGFR2 from cell lysates and incubated the immunoprecipitated receptor with recombinant active ERK1 and radioactive [γ-^32^P]-labelled adenosine triphosphate (ATP). The experiment was performed in the presence of PD173074 (FGFR inhibitor) to prevent autophosphorylation of the receptor. Using autoradiography, we could observe a band representing phosphorylated FGFR2 in the presence of active ERK1 (Figure 4d). This band was somewhat reduced in the sample from FGFR2 S780A cells. Thus, it seems that ERK1 directly phosphorylates FGFR2 on S780A. Since the phosphorylation of FGFR2 S780A is only partially reduced, we cannot exclude that other sites in FGFR2 might be phosphorylated by ERK1.

In order to study the role of S780 in FGFR2 further, we also prepared cell lines stably expressing FGFR2 S780D. The negatively charged aspartic acid might mimic constitutive phosphorylation of the residue. First, we verified that U2OS-R2 S780D cells were able to activate the main signaling pathways as wild-type FGFR2 (Appendix A). We then investigated the tyrosine phosphorylation levels of FGFR2 upon FGF1 stimulation in U2OS-R2 S780D cells. Unfortunately, we observed the same effect of serine 780 mutated to an aspartic acid as we observed for FGFR2 S780A (Appendix A). This is not surprising, as the mimicry of a phosphorylated serine by an aspartic acid often fails to reproduce the function of the phosphorylated serine [19]. We think that, instead of mimicking a constitutively phosphorylated serine, FGFR2 S780D rather displays a site that has lost its ability to become phosphorylated. Thus, FGFR2 S780D acts similarly to FGFR2 S780A and shows increased FGFR2 tyrosine phosphorylation.

### 3.3. Possible Role of Serine 780 in FGFR2 in Cancer Progression

We next investigated if the ERK1/2-mediated negative feedback loop possibly could play a role in cancer progression. By exploring databases reporting known alterations in cancer (cBioPortal; http://www.cbioportal.org and COSMIC (Catalogue of Somatic Mutations in Cancer); http://cancer.sanger.ac.uk) [20,21,22], we found several alterations that might influence the negative feedback loop in FGFR2 (Figure 5a). First, several truncated versions of FGFR2 lacking the C-terminal tail, including S780, have been identified in thyroid, skin, endometrial, and gastric cancers. In these cases, the negative feedback loop will not be operational and the receptor signaling may not be properly attenuated. This could potentially contribute to cancer progression. Secondly, several mutations in the close proximity of S780 have also been identified, including the glutamic acid at position 777 to a lysine and tyrosine 779 to a cysteine (Figure 5a). It is possible that these mutations influence S780 phosphorylation and receptor activity. Especially, the exchange of a negatively charged glutamic acid to a positively charged lysine might affect the properties of this region. Interestingly, serine 780 in FGFR2 has been found mutated to leucine in a patient with bladder cancer (Figure 5a). We decided to investigate the effect of this mutation further.

First of all, we generated U2OS cells stably expressing FGFR2 S780L (U2OS-R2 S780L) and confirmed that FGFR2 S780L cells were able to activate signaling pathways similarly to wild-type FGFR2 (Figure 5b and Appendix A). Moreover, the levels of FGFR2 wild-type and FGFR2 S780L seem comparable and the mutated receptor is able to bind FGF1 at the cell surface and internalize FGF1 into early endosomes similarly to wild-type FGFR2 (Figure 5c and Appendix A). In addition, the internalized DL550-FGF1 co-localizes well with anti-FGFR2 staining (Figure 5d).

We then analyzed the levels of FGFR2 tyrosine phosphorylation over time in FGFR2 S780L-expressing cells stimulated with FGF1. Compared to wild-type expressing cells, FGFR2 activation was prolonged in U2OS-R2 S780L (Figure 6a). Similar results were observed in two additional clones of U2OS-R2 S780L (Appendix A).

Clearly, the mutation of serine 780 to leucine leads to increased receptor signaling, which may be an advantage for cancer cells. Most cancer deaths (~90%) are caused by metastasis [23]. In order to metastasize and spread to distant organs, cancer cells need to be mobile and able to migrate. We therefore tested the mobility of U2OS cells stably expressing wild-type or S780 mutants. Since clonal variations might occur, we tested three different clones of each. Cells were seeded sparsely to allow for random migration and then imaged every 10 min for 21 h. We observed that stimulation of cells with FGF1 increased the migration velocities of all cell lines (Figure 6b and Appendix A). Moreover, U2OS cells expressing either of the mutant forms of S780 (A/L) migrated significantly faster than wild-type expressing cells in the presence of FGF1 (Figure 6b and Appendix A). Preventing the negative feedback loop in FGFR2 by mutation of S780 causes increased signaling and, as a consequence, increased cell migration. In a cancer setting, this might contribute to disease progression. 

## 4. Discussion

We have identified a negative feedback loop, mediated by the ERK1/2 pathway that regulates FGFR2 signaling. First, we found that inhibition of the ERK1/2-pathway leads to sustained FGFR2 signaling. Next, we found that substituting serine 780 in FGFR2 with alanine or leucine results in increased signaling. Serine 780 in FGFR2 is followed by a proline and thus forms an ERK1/2 phosphorylation motif (pS/T-P). In addition, S780 in FGFR2 corresponds to S777 in FGFR1. S777 in FGFR1 has previously been shown to be phosphorylated directly by ERK1/2. Taken together, we propose that ERK1/2-mediated phosphorylation of S780 in FGFR2 acts as a negative feedback loop to prevent excess signaling. This was evident when cells were pretreated with EGF to activate ERK1/2 prior to FGF1 stimulation. In activated cells, the response to FGF1 was lower than in resting cells. The feedback loop may function to fine-tune FGFR2 signaling in an environment where signaling is already on, preventing a further increase in signaling. We observed that cells lacking S780 (mutated to alanine or leucine) migrate faster than wild-type expressing cells. Since migration is important for spreading of cancer cells and metastasis, clearly the lack of this negative feedback loop gives cancer cells an advantage. Indeed, S780L has been identified in a patient with bladder cancer. In addition, several truncated forms of FGFR2 lacking S780 have also been identified in cancer. Maintaining the negative feedback loop ensures accurate signaling and preventing the feedback loop (either by mutation of S780 in FGFR2 or by inhibition of ERK1/2 signaling) could cause cancer progression.

Aberrant signaling through the Ras-Raf-MEK1/2-ERK1/2 pathway has been implicated in many types of cancer and is a promising therapeutic target. Although BRaf- and MEK-inhibitor mono- or combination-therapy have shown promising effects in cancer patients, many patients develop resistance and experience disease progression [24]. These resistance mechanisms include reactivation of the MAPK and/or the PI3K/Akt pathway. Examples of common resistance mechanisms include NRas mutation, BRaf v600 amplification, loss of PTEN, PI3KCA mutation, and RTK activation [24]. Since ERK1/2 is the only activator in the pathway with the ability to stimulate a wide variety of downstream substrates, it has emerged as an attractive therapeutic target. Despite the discovery of ERK1/2 many decades ago, ERK1/2 inhibitors have so far not been successfully implemented in the clinic [25]. One possible reason is that activated ERK1/2 stimulates inhibitory phosphorylation of many upstream factors and kinases, such as MEK, Raf, and different RTKs (including FGFR2), which prevent extensive signaling [25,26]. It is therefore worth considering that sole inhibition of the ERK1/2 signaling pathway in cancer could give rise to increased FGFR signaling through other signaling pathways (for example PI3K/Akt). Indeed, a recent study showed an increase in FGFR signaling upon MEK inhibition in KRas-driven lung cancer [27]. Therefore, caution should be taken when considering the use of MEK/ERK pathway inhibitors in cancer patients with FGFR2 expression.

Interestingly, a similar negative feedback loop involving ERK1/2-mediated phosphorylation has been identified for EGFR. In this case, ERK1/2-mediated signaling phosphorylates EGFR at threonine 669 (T669) [28,29]. Phosphorylation of this residue, which is localized in the juxtamembrane region of the receptor, was shown to reduce the tyrosine phosphorylation levels of EGFR. It seems that the T669-phosphorylated juxtamembrane region in EGFR has a reduced ability to cross-activate the other receptor of the dimer [30]. Although S780 is located in the C-terminal tail of FGFR2, it is possible that local conformational changes introduced by the phosphorylation at S780 reduce its cross-activation. It is also possible that a local conformational change in the receptor, caused by phosphorylation of S780, makes FGFR2 a better substrate for tyrosine dephosphorylation. Another attractive possibility is that phosphorylated S780 directly recruits a negative regulator such as a phosphatase or a scaffolding protein. Interestingly, serine 779 (S782 according to our numbering) in FGFR2 is phosphorylated by active PKCε and provides a docking site for the adaptor protein 14-3-3 [31]. However, in this case, phosphorylation of S779 (S782) seems to be required for sustained ERK1/2 activation and thus does not function as a negative feedback loop. It will be interesting to understand how these two phosphorylation events at S779 (S782) and S780 in FGFR2 work in partnership. It is also possible that S780 phosphorylation plays a role in receptor endocytosis and degradation. Previously, we found that S789 in FGFR1 is phosphorylated by RSK2 and seems to be required for proper internalization [13]. Interestingly, when we treated FGF1 stimulated cells with U0126 and stained for p-FGFR, we observed increased p-FGFR staining close to the cell surface (Figure 1d). We also observed less FGF1 internalized. However, when U2OS-R2 S780A cells were treated with U0126, the uptake of DL550-FGF1 was reduced to a similar extent as in U2OS-R2 wild-type cells (Figure 4c). It is possible that an RSK2-mediated feedback loop similar to that observed for FGFR1 exists also for FGFR2. U0126 inhibits both ERK1/2 and its downstream target, RSK2. There are also other examples of receptors that are serine-phosphorylated similarly to FGFR2, FGFR1, and EGFR. The Met receptor is phosphorylated by active PKCδ/ε at serine 985 in the juxtamembrane region [32]. Substitution of serine 985 by alanine resulted in increased tyrosine phosphorylation of Met. Similarly to FGFR2 and FGFR1, it is not clear what causes the reduced tyrosine phosphorylation in this case. Taken together, serine and threonine phosphorylation of receptor tyrosine kinases might be a common event that regulates receptor activity. A better understanding of these events will provide useful information when targeting receptor tyrosine kinases in cancer.

Although the mutation of serine 780 in FGFR2 to leucine clearly increases FGFR2 tyrosine phosphorylation levels and FGF1-stimulated cell migration, the role of this mutation in cancer is not clear. The mutation was found in a patient with bladder cancer. Although increased signaling and increased migration are traits that normally would benefit cancer cells, the role of FGFR2 signaling in bladder cancer is not fully understood and the FGFR2b isoform has been suggested to act as a tumor suppressor in the urothelium. It has been reported that reduction of FGFR2b levels in urothelial cancer samples correlate with decreased survival [33] and the chromosomal arm 10q, where the *FGFR2* gene is located, is often lost in advanced bladder cancer [34]. It should be noted that 10q also contains the tumor suppressor *PTEN* [35]. Moreover, expression of FGFR2b in urothelial cells lacking endogenous FGFR2 led to reduced proliferation and reduced tumorigenicity in nude mice [36]. On the other hand, increased FGFR2c expression has been reported in a model of epithelial-to-mesechymal transition (EMT) in bladder cancer cells [37] and recently, the U.S. Food and Drug Administration (FDA) approved Balversa (Erdafitinib), a pan FGFR-inhibitor for clinical use in patients with locally advanced or metastatic urothelial carcinoma with FGFR2 and FGFR3 aberrations [38]. This is the first targeted-FGFR therapy approved for clinical use and the first targeted therapy in advanced urothelial carcinoma. Alongside the approval of the drug, the FDA approved an RT-PCR-based diagnostic test to identify patients with FGFR3 mutations or FGFR2 fusions. It is possible that FGFR2 plays a tumor-suppressing role in earlier stages of bladder cancer, but could have tumor-promoting effects in certain patients with advanced bladder cancer.

The S780L mutation is only reported once in the COSMIC database and at such low frequency that the significance is questionable. On the other hand, truncated versions of FGFR2 lacking S780 have been identified in cancer patients (Figure 5a). In addition, an alternatively spliced form of FGFR2, FGFR2IIIb-C3, is also lacking S780 [39]. In contrast to full-length FGFR2IIIb, FGFR2IIIb-C3 is only identified in human cancer samples. Aberrant expression of FGFR2IIIb-C3 in SUM-52 breast cancer cells resulted in sustained signaling leading to transformation [40]. A tyrosine phosphorylation site (corresponding to Y766 in FGFR1) is also lacking in FGFR2IIIb-C3 and could explain the increased signaling and transformation capabilities of FGFR2IIIb-C3. However, mutation of only this tyrosine in full-length FGFR2 did not lead to increased signaling [41]. Thus, loss of other mechanisms maintained by the C-terminal tail of FGFR2 might cause the increased signaling and transforming potential of FGFR2IIIb-C3. We propose that lack of S780 in FGFR2IIIb-C3 could promote its transforming capabilities. Interestingly, a patient with endometrial cancer was identified with the activating mutation N549H in FGFR2 and a truncated C-terminal tail (cBioPortal; http://www.cbioportal.org) [20,21]. This combination of alterations in FGFR2 will clearly impact signaling output and could be even more cancer-promoting than versions with either mutation alone.

In summary, we have identified an ERK1/2-mediated negative feedback-loop in FGFR2. We propose that lack of this feedback loop could give cancer cells an advantage and, indeed, variants of FGFR2 lacking the feedback loop have been identified in several human cancers. We conclude therefore that, in addition to the previously reported activating mutations in the kinase domain of FGFRs [4], mutations in the C-terminal tail of FGFR2 may also cause hyperactivation of the receptors.

## Figures and Tables

**Figure 1 cells-08-00518-f001:**
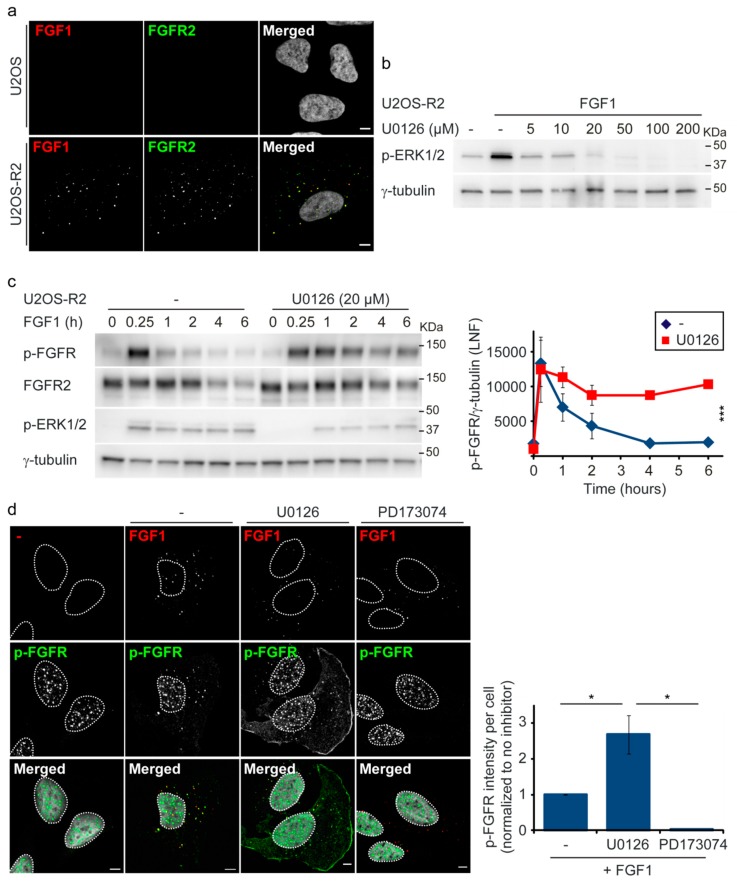
Inhibition of the ERK1/2 pathway prolongs FGFR2 signaling. (**a**) U2OS cells or U2OS cells stably transfected with FGFR2IIIc (U2OS-R2) were treated with 200 ng/mL DL550-FGF1 in the presence of heparin (50 U/mL) for 30 min. The cells were then fixed, stained with anti-FGFR2 antibodies and Hoechst, and analyzed by confocal microscopy. The images were taken at fixed intensity settings, and brightness/contrast was adjusted in the same way for all images. Representative images are shown. Scale bar: 5 µM. (**b**) U2OS-R2 cells were kept in serum-free media for 2 h prior to stimulation for 30 min with 200 ng/mL FGF1 in the presence of heparin (20 U/mL) and increasing concentrations of U0126. Cells were then lysed and the lysates were analyzed by immunoblotting using the indicated antibodies. A p in front of the name of the antibody indicates that it recognizes the phosphorylated form of the protein. One representative experiment is shown. (**c**) U2OS-R2 cells were kept in serum-free media for 2 h before addition of 100 ng/mL FGF1 and heparin (20 U/mL) in the presence or absence of U0126 (20 µM) for indicated periods of time. Cycloheximide (10 µg/mL) was added at the beginning of the starvation period and kept throughout the experiment. After lysis, the cellular material was analyzed with immunoblotting using the indicated antibodies. A p in front of the name of the antibody indicates that it recognizes the phosphorylated form of the protein. Quantifications of three independent experiments are presented in the graph. The bands corresponding to phosphorylated receptor were normalized to Lane normalization factor (LNF) (γ-tubulin). Error bars denote the standard deviation. The difference between U0126 treated cells versus untreated cells was significant (*p* ≤ 0.001, 3-way ANOVA, Holm-Sidak test, n = 3). (**d**) U2OS-R2 cells were kept in serum-free media for 2 h before addition of 200 ng/mL DL550-FGF1 and heparin (50 U/mL) for 30 min. The cells were pretreated with U0126 (20 µM) or PD173074 (50 nM) 30 min before addition of FGF1 as indicated. The cells were then fixed, stained with anti-p-FGFR antibodies and Hoechst, and analyzed by confocal microscopy. Scale bar: 5 µM. Quantifications of two independent experiments were performed as described in materials and methods and are presented in the graph. In total, 60 cells treated with FGF1 alone, 60 cells treated with FGF1 and U0126, and 38 cells treated with FGF1 and PD173074 were quantified. Outliers were removed according to the 1.5*IQR outlier rule. Error bars denote the standard error of the mean (SEM), n = 2. Due to a general variation in the intensity between the two experiments, the means in each experiment were normalized to the mean of cells treated with FGF1 alone (no inhibitor) in the corresponding experiment. (* *p* ≤ 0.05, two-sided t test on normalized data, n = 2).

**Figure 2 cells-08-00518-f002:**
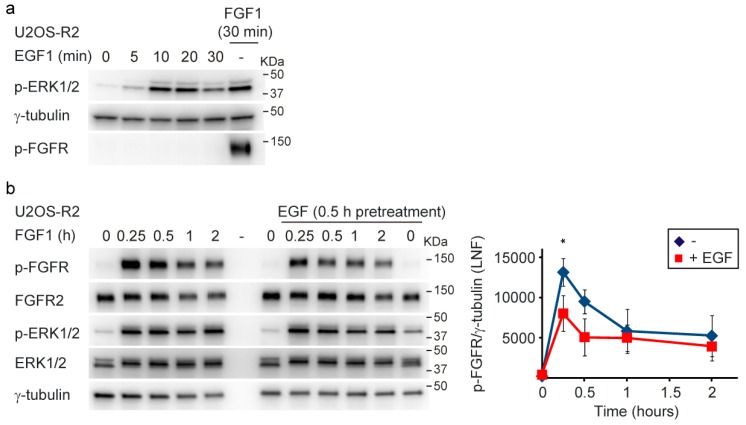
Pretreatment with EGF reduces the response to FGF1. (**a**) U2OS-R2 cells were kept in serum-free media for 2 h and then 100 ng/mL EGF or 100 ng/mL FGF1 was added to the cells. The cells were lysed after the indicated periods of time. A p in front of the name of the antibody indicates that it recognizes the phosphorylated form of the protein. (**b**) U2OS-R2 cells were kept in serum-free media for 2 h. Then, EGF (20–100 ng/mL) was added to the samples as indicated. After 30 min, the cells were stimulated with 20 ng/mL FGF1 and heparin (10 U/mL) and lysed after the indicated periods of time. Cycloheximide (10 µg/mL) was added at the beginning of the starvation period and kept throughout the experiment. After lysis, the cellular material was analyzed by immunoblotting using the indicated antibodies. A p in front of the name of the antibody indicates that it recognizes the phosphorylated form of the protein. Quantifications of three independent experiments are presented in the graph. The time-point of 30 min is only from two experiments. The bands corresponding to phosphorylated receptor were normalized to LNF (γ-tubulin). Error bars denote the standard deviation. The difference between EGF-pretreated cells versus cells not treated with EGF was significant at the time point of 15 min (* *p* ≤ 0.05, 1-way ANOVA, Tukey test, n = 3).

**Figure 3 cells-08-00518-f003:**
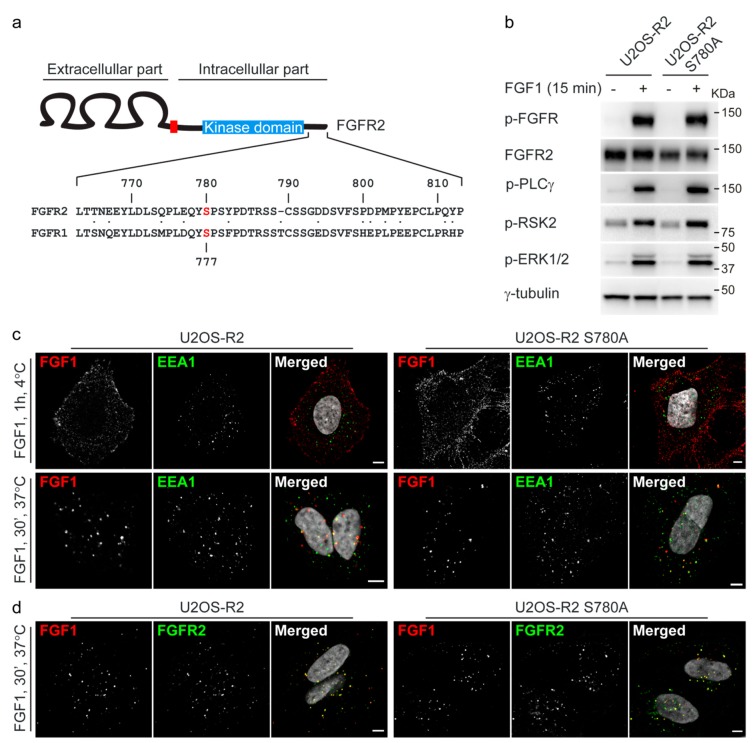
Characterization of cell lines stably expressing FGFR2 S780A mutant. (**a**) A pairwise sequence alignment tool from EMBL-EBI was used to align the C-terminal tails of FGFR2 and FGFR1. S780 in FGFR2 corresponds to S777 in FGFR1 (labelled in red in the figure). Numbers refer to the amino acid numbering used for human FGFR2 (NCBI: NM_000141). (**b**) U2OS-R2 cells or U2OS-R2 S780A cells were kept in serum-free media for two hours and then treated or not with 100 ng/mL FGF1 for 15 min in the presence of heparin (20 U/mL). After lysis, the cellular material was analyzed by immunoblotting using the indicated antibodies. A p in front of the name of the antibody indicates that it recognizes the phosphorylated form of the protein. One representative experiment is shown. (**c**) U2OS-R2 or U2OS-R2 S780A cells were kept at 4 °C with DL550-FGF1 for one hour in the presence of heparin (50 U/mL). Next, the cells were either fixed directly (upper panel) or incubated for 30 min at 37 °C before fixation (lower panel). The cells were then stained with anti-EEA1 antibodies and Hoechst and analyzed by confocal microscopy. Representative images are shown. Scale bar: 5 µM. (**d**) U2OS-R2 or U2OS-R2 S780A cells were treated with 200 ng/mL DL550-FGF1 in the presence of heparin (50 U/mL) for 30 min. The cells were then fixed, stained with anti-FGFR2 antibodies and Hoechst, and analyzed by confocal microscopy. Representative images are shown. Scale bar: 5 µM.

**Figure 4 cells-08-00518-f004:**
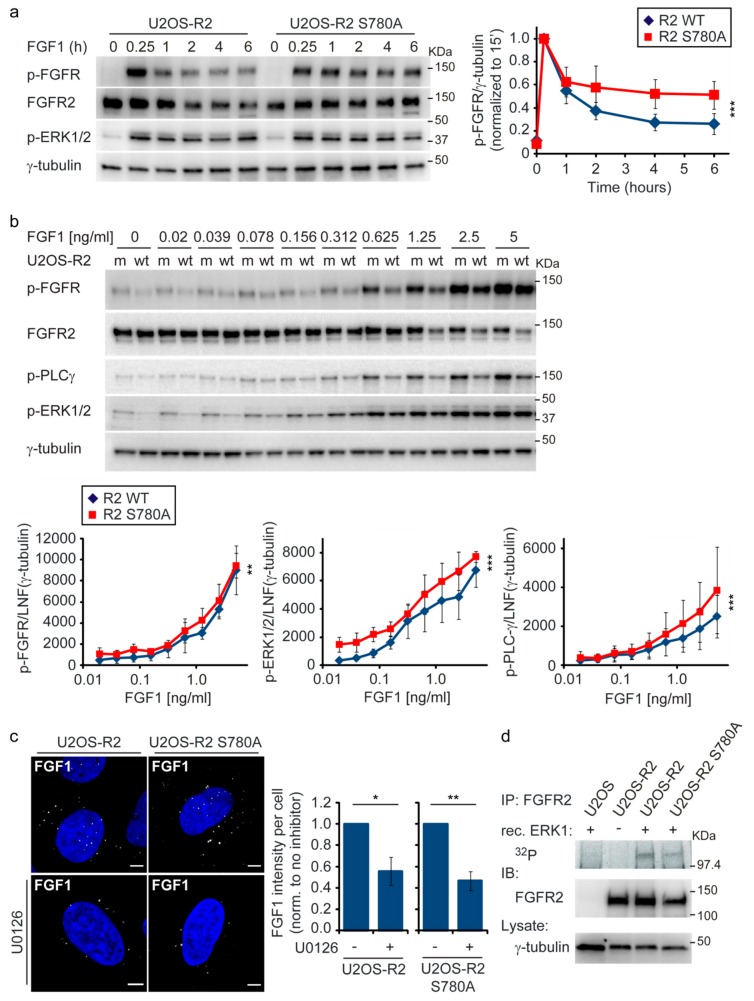
Signaling from FGFR2 S780A is prolonged compared to wild-type FGFR2. (**a**) U2OS-R2 wild-type or U2OS-R2 S780A cells were kept for two hours in serum-free media before addition of 100 ng/mL FGF1 and heparin (20 U/mL) for the indicated periods of time. Cycloheximide (10 µg/mL) was added at the beginning of the starvation period and kept throughout the experiment. After lysis, the cellular material was analyzed by immunoblotting using the indicated antibodies. A p in front of the name of the antibody indicates that it recognizes the phosphorylated form of the protein. Quantifications of five independent experiments are presented in the graph. The bands corresponding to phosphorylated receptor were normalized to γ-tubulin and within each experiment to the time point of 15 min. Error bars denote the standard deviation. The difference between FGFR2 wild-type and the S780A mutant was significant (*p* ≤ 0.001, 3-way ANOVA, Holm-Sidak test, n = 5). (**b**) U2OS-R2 wild-type (wt) or U2OS-R2 S780A (m) cells were kept for two hours in serum-free media before addition of indicated concentrations of FGF1 in the presence of heparin (20 U/mL) for 15 min. After lysis, the cellular material was analyzed by immunoblotting using the indicated antibodies. A p in front of the name of the antibody indicates that it recognizes the phosphorylated form of the protein. Quantifications of three independent experiments are presented in the graph. The bands corresponding to the phosphorylated receptor/ERK1/2/PLCγ were normalized to LNF (γ-tubulin). Error bars denote the standard deviation. The difference between FGFR2 wild-type and S780A mutant was significant (** *p* ≤ 0.01, *** *p* ≤ 0.001, 3-way ANOVA, Holm-Sidak test, n = 3). (**c**) Internalization of DL550-FGF1 in FGFR2 and FGFR2 S780A cells is reduced upon U0126 treatment. U2OS-R2 and U2OS-R2 S780A cells were incubated with 200 ng/mL DL550-FGF1 and heparin (50 U/mL) for 30 min. The cells were pretreated as indicated with U0126 (20 µM) for 30 min before addition of FGF1. The cells were then fixed, stained with Hoechst, and analyzed by confocal microscopy. The images were taken at fixed intensity settings, and brightness/contrast was adjusted in the same way for all images. Scale bar: 5 µM. Quantifications of four independent experiments were performed as described in materials and methods and are presented in the graph. In total, 269 U2OS-R2 cells, 262 U2OS-R2 cells treated with U0126, 247 U2OS-R2 S780A cells, and 213 U2OS-R2 S780A cells treated with U0126 were quantified. Outliers were removed according to the 1.5*IQR outlier rule. Error bars denote the SEM (n = 4). Due to a general variation in the intensity between experiments, the means of U0126 treated cells for each cell line in each experiment were normalized to the mean of the corresponding cell line in the same experiment (** *p* ≤ 0.05, * *p* ≤ 0.05, two-sided t test, n = 4). (**d**) In vitro phosphorylation of FGFR2 by active recombinant ERK1. Lysates from U2OS, U2OS-R2, and U2OS-R2 S780A cells were subjected to FGFR2 immunoprecipitation (IP). The immunoprecipitated materials were next incubated with [γ-^32^P]-labelled adenosine triphosphate and recombinant active ERK1 (rec. ERK1) in the presence of PD173074. After washing, the samples were subjected to SDS-PAGE and analyzed by autoradiography and immunoblotting (IB). One representative experiment is shown.

**Figure 5 cells-08-00518-f005:**
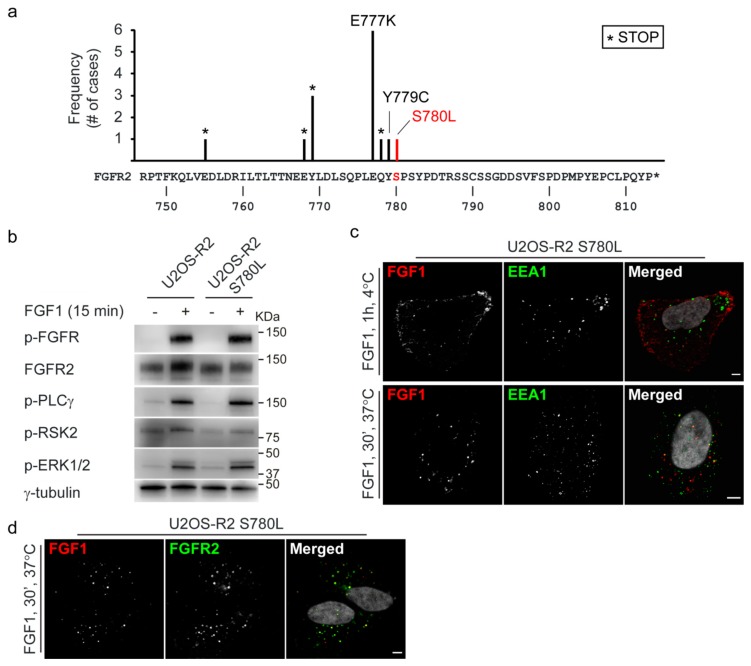
FGFR2 S780 is mutated in cancer. (**a**) The sequence of the C-terminal tail of FGFR2. Variations that might influence S780 phosphorylation and have been identified in cancer patients are indicated. An asterisk indicates a stop codon. Numbers refer to the amino acid numbering used for human FGFR2 (NCBI: NM_000141). The variations are reported in cBioPortal and COSMIC. **(b)** U2OS-R2 cells or U2OS-R2 S780L cells were kept in serum-free media for two hours and then treated or not with 100 ng/mL FGF1 for 15 min in the presence of heparin (20 U/mL). After lysis, the cellular material was analyzed by immunoblotting using the indicated antibodies. A p in front of the name of the antibody indicates that it recognizes the phosphorylated form of the protein. One representative experiment is shown. (**c**) U2OS-R2 S780L cells were kept at 4 °C with DL550-FGF1 for one hour. Next, the cells were either fixed directly (upper panel) or incubated for 30 min at 37 °C before fixation (lower panel). The cells were then stained with anti-EEA1 antibodies and Hoechst and analyzed by confocal microscopy. Representative images are shown. Scale bar: 5 µM. (**d**) U2OS-R2 S780L cells were treated with 200 ng/mL DL550-FGF1 in the presence of heparin (50 U/mL) for 30 min. The cells were then fixed, stained with anti-FGFR2 antibodies and Hoechst, and analyzed by confocal microscopy. Representative images are shown. Scale bar: 5 µM.

**Figure 6 cells-08-00518-f006:**
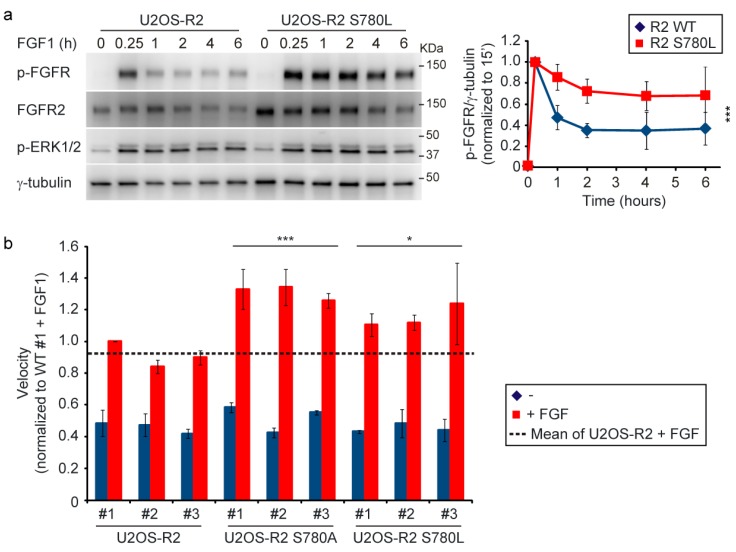
Lack of S780 phosphorylation increases the biological response to FGF1. (**a**) U2OS-R2 wild-type or U2OS-R2 S780L cells were kept in serum-free media for two hours before addition of 100 ng/mL FGF1 and heparin (20 U/mL) for indicated periods of time. Cycloheximide (10 µg/mL) was added at the beginning of the starvation period and kept throughout the experiment. After lysis, the cellular material was analyzed by immunoblotting using the indicated antibodies. A p in front of the name of the antibody indicates that it recognizes the phosphorylated form of the protein. Quantifications of three independent experiments are presented in the graph. The bands corresponding to phosphorylated receptor were normalized to γ-tubulin and within each experiment to the time point of 15 min. Error bars denote the standard deviation. The difference between FGFR2 wild-type and S780L mutant was significant (*p* ≤ 0.001, 3-way ANOVA, Holm-Sidak test, n = 3). (**b**) Three different clones of U2OS-R2 wild-type, U2OS-R2 S780A, and U2OS-S780L cells were seeded into Image Lock 96-well plates. The cells were left untreated or stimulated with FGF1 (100 ng/mL) in the presence of heparin (20 U/mL) and imaged every 10 min over a period of 21 h by IncuCyte^®^ S3 Live Cell Analysis System. The graph represents the mean velocities normalized to U2OS-R2 clone #1 with FGF1 of three independent experiments. The total number of cells tracked: U2OS-R2 #1 (-/+): 138/140, #2 (-/+): 111/138, #3 (-/+): 106/141, U2OS-R2 S780A #1 (-/+): 139/167, #2 (-/+): 135/142, #3 (-/+): 128/142, U2OS-R2 S780L #1 (-/+): 97/174, #2 (-/+): 110/153, #3 (-/+): 121/139. Error bars denote the SEM (n = 3). The difference between the wild-type and S780 mutants were significant (*** *p* ≤ 0.001, * *p* ≤ 0.05, student t test, n = 3).

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
