# Peer review of "Cancer Mutations in FGFR2 Prevent a Negative Feedback Loop Mediated by the ERK1/2 Pathway"

_cells, 2019, doi:10.3390/cells8060518_

Round 1

Reviewer 1 Report

This is well written and clear manuscript showing that S780 in FGFR2 has a role in a negative feedback loop leading to receptor de-phosphorylation after activation and therefore regulating receptor activity. The authors have shown a similar mechanism in FGFR1 therefore the idea behind the study is not completely novel. However, the results are relevant and of interest.

In the first part of the study the authors investigate the effect of S780 alteration on FGFR2 phosphorylation and signalling in detail and present robust and persuasive data to support their theory. However, the second part of the study looking at a possible role in cancer is weaker. The COSMIC catalogue is used to look for evidence of mutations of this residue in cancer. However, only one mutation at position S780 of FGFR2 is found in this database, which contains information on over 72,000 tumours. Such a low frequency raises questions on the significance of a role in cancer. Moreover, this single mutation is found in a bladder cancer patient. There is no much evidence that activation of FGFR2 plays a role in bladder cancer. In fact some studies have shown that FGFR2 may, on the contrary, have a protective role against bladder cancer (for example see D. Ricol, D. Cappellen, A. El Marjou et al., “Tumour suppressive properties of fibroblast growth factor receptor 2-IIIb in human bladder cancer,” Oncogene, vol. 18, no. 51, pp. 7234–7243, 1999). Therefore it is unclear how a mutation that potentially lead to a prolonged activation of FGFR2 could contribute to bladder cancer progression. In my opinion the authors should either provide more evidence supporting the link with cancer in general and bladder in particular, or discuss these limitations of the study in details. 

Overall the manuscript has merit and I recommend publication pending modifications.

Author Response

Reviewer 1:

We thank the reviewer for acknowledging the relevance of our findings and for the valuable feedback.

In the first part of the study the authors investigate the effect of S780 alteration on FGFR2 phosphorylation and signalling in detail and present robust and persuasive data to support their theory. However, the second part of the study looking at a possible role in cancer is weaker. The COSMIC catalogue is used to look for evidence of mutations of this residue in cancer. However, only one mutation at position S780 of FGFR2 is found in this database, which contains information on over 72,000 tumours. Such a low frequency raises questions on the significance of a role in cancer. Moreover, this single mutation is found in a bladder cancer patient. There is no much evidence that activation of FGFR2 plays a role in bladder cancer. In fact some studies have shown that FGFR2 may, on the contrary, have a protective role against bladder cancer (for example see D. Ricol, D. Cappellen, A. El Marjou et al., “Tumour suppressive properties of fibroblast growth factor receptor 2-IIIb in human bladder cancer,” Oncogene, vol. 18, no. 51, pp. 7234–7243, 1999). Therefore it is unclear how a mutation that potentially lead to a prolonged activation of FGFR2 could contribute to bladder cancer progression. In my opinion the authors should either provide more evidence supporting the link with cancer in general and bladder in particular, or discuss these limitations of the study in details.

We agree with the reviewer that this is an important point that should be discussed further. We have now included a more balanced discussion on the role of the mutation in FGFR2 in cancer and particularly in bladder cancer (see page 15). We thank the reviewer for pointing this out.

Reviewer 2 Report

The Manuscript by Szybowska et al ., demonstrates an elegant dissection of the feedback loop between FGFR2 and ERK signalling. They provide a logical, clear progression of ideas which clearly correlate with their conclusions and are supported by their data. It is a valuable contribution to the body of work generated by this team, builds on their previous findings and has value to pre-clinical work targeting FGFRs. The authors should be commended for their transparency of approach- particularly with difficult reagents that are in the FGFR field- i.e acknowledgement that the nuclear pFGFR signal is an artefact. 

I highly recommend for acceptance with minor issues that I would like clarified. 

Densitometry- field standard for analysis of immunoblots typically corrects Phospho to total abundance of a particular protein, then corrects that value to loading (i.e tubulin).  i.e figure 1 should have a total ERK blot. with pERK corrected to tERK, normalised to g-tubulin. 

Images of the U2OSR2 (and all mutant lines) stained for at least FGFR2, (as they have the antibodies, FGFR1-4 immunofluorescence would also be ideal).

Is there any differences in receptor surface localisation between wild-type and mutant receptors? i.e are they exhibiting extended signalling due to alterations in receptor internalization dynamics? There are many imaging and biochemical strategies available to test this, and it would strengthen their observations if the issue is the mutant requires MEK-dependent phosphorylation to regulate surface internalization and attenuation of the signalling.  

Could authors differentiate between FGFR2c and FGFR2b? It is clear from the ligand, but might not be to the non-field expert.

Author Response

Reviewer 2:

We thank the reviewer for the kind remarks and the valuable comments and suggestions.

Densitometry- field standard for analysis of immunoblots typically corrects Phospho to total abundance of a particular protein, then corrects that value to loading (i.e tubulin).  i.e figure 1 should have a total ERK blot. with pERK corrected to tERK, normalised to g-tubulin.

We agree with the reviewer that phospho- to total abundance of a particular protein is the optimal way to quantify phosphorylated proteins. By doing it this way, it is possible to distinguish between an actual alteration in the phosphorylation level of the protein of interest and a potential alteration in the protein level of the particular phospho-protein. Here, we normalize against loading control (γ-tubulin) instead and we can only claim that we see an alteration in the level of phosphorylated protein and not necessarily in the level of protein phosphorylation. The text is now altered to be more precise on this point.

Images of the U2OSR2 (and all mutant lines) stained for at least FGFR2, (as they have the antibodies, FGFR1-4 immunofluorescence would also be ideal).

We have now included images of U2OS and U2OS-R2 wt, A and L mutant cell lines showing anti-FGFR2 staining (Figure 1a, 3d and 5d). In addition we have included images with anti-FGFR1-4 staining in U2OS cells (Figure S1a). In untransfected U2OS cells, as expected, none of the FGFRs are detected by antibody staining. We have therefore included as a positive control, transient overexpression of the different FGFRs to ensure that the staining worked well in our experiments.

Is there any differences in receptor surface localisation between wild-type and mutant receptors? i.e are they exhibiting extended signalling due to alterations in receptor internalization dynamics? There are many imaging and biochemical strategies available to test this, and it would strengthen their observations if the issue is the mutant requires MEK-dependent phosphorylation to regulate surface internalization and attenuation of the signalling.

This is an interesting point and we have now performed several experiments to investigate this further. First of all, in figure 1d, we could detect more surface staining in cells treated with U0126 (MEK1/2 inhibitor). This indicates a decrease in endocytosis when MEK1/2-ERK1/2 signalling is inhibited. However, from our work on FGFR1, we know that even though we observed this decrease in endocytosis upon MEK1/2-ERK1/2 inhibition, this effect was not due to lack of ERK1/2 phosphorylation of the receptor but rather a lack of a second serine phosphorylation event in FGFR1 mediated by RSK2 (Nadratowska-Wesolowska et al. 2014). To investigate this, we compared the uptake of DL550-FGF1 in U2OS-R2 wt and U2OS-R2 S780A mutant cells in the presence of U0126 (Figure 4c). When FGFR2 wt and FGFR2 S780A expressing cells were treated with U0126, the uptake of DL550-FGF1 was reduced similarly in both cell lines. This means that lack of phosphorylation on S780 is not the reason for the reduced endocytosis upon MEK1/2 inhibiton. We also performed in vitro phosphorylation experiments to show that ERK directly phosphorylates FGFR2 at this position (Figure 4d). Taken together, we suspect that a similar mechanism as in the case of FGFR1 exists for FGFR2, and that other components of the MAPK pathway (such as RSK2?) might also phosphorylate FGFR2. This phosphorylation event might be important for endocytosis. This still have to be investigated. However, despite our great interest in investigating this further, we believe that this is beyond the scope of this paper.

Could authors differentiate between FGFR2c and FGFR2b? It is clear from the ligand, but might not be to the non-field expert.

Thanks for raising this question. We have now emphasised in the manuscript that we here have studied isoform FGFR2 IIIc (see first paragraph in the Result part and 2.2. Plasmids and siRNAs in the Material and methods).